# Dietary Adherence and Cognitive Performance in Older Adults by Nativity Status: Results from the National Health and Nutrition Examination Survey (NHANES), 2011–2014

**DOI:** 10.3390/geriatrics9020025

**Published:** 2024-02-25

**Authors:** Md Towfiqul Alam, Elizabeth Vásquez, Jennifer L. Etnier, Sandra Echeverria

**Affiliations:** 1Department of Health Sciences, James Madison University, Harrisonburg, VA 22807, USA; 2Department of Epidemiology & Biostatistics, University at Albany, Albany, NY 12144, USA; evasquez2@albany.edu; 3Department of Kinesiology, University of North Carolina at Greensboro, Greensboro, NC 27412, USA; jletnier@uncg.edu; 4Department of Public Health Education, University of North Carolina at Greensboro, Greensboro, NC 27412, USA; seecheve@uncg.edu

**Keywords:** dietary adherence, cognitive performance, NHANES, older adults, nativity status

## Abstract

Although adherence to dietary guidelines is associated with better cognitive performance, there may be differences by nativity status. This study aimed to investigate the association between adherence to the healthy eating index (HEI) and cognitive performance (CP) among United States (US)-born and foreign-born older adults (60+ years). Data were obtained from the 2011–2014 NHANES (*n* = 3065). Dietary adherence was assessed with HEI quintiles. CP (adequate vs. low) was examined using word listing (CERAD), animal naming (AFT), and the digit symbol substitution test (DSST). Weighted multivariable logistic regressions were used to examine associations. The US-born participants in higher dietary quintiles had adequate CP when compared to foreign-born participants. In adjusted models, the US-born participants in the highest HEI quintile had increased odds of adequate DSST scores (odds ratio: 1.95, 95% confidence interval: 1.15–3.28) compared with those in the lowest quintile. Patterns of association were generally reversed for foreign-born participants and were non-statistically significant. Future research should consider the influence of diets in delaying or preventing decline in cognition and evaluate nutritional factors that contribute to cognitive outcomes for the foreign-born population.

## 1. Introduction

A central concern when evaluating cognition is whether cognitive performance (CP) in the form of impairments and cognitive decline are related to aging-specific disease processes or are reflective of broader social determinants of health, such as belonging to a minority racial/ethnic group or nativity status. This is an important issue given the rapid growth of the older adult populations and their burden of diseases and risk factors related to CP, such as vascular risk factors including hypertension and diabetes, low levels of education, and excess body weight [1].

An abundant body of work has shown that, in addition to physical activity, a heart healthy diet comprised of adequate amounts of plant food (fruits, vegetables, legumes, cereals) and fish, as well as low amounts of dairy, red meats, sugar, and alcohol content, maintains or enhances CP [2,3,4,5]. Prior research has focused on examining the role of individual food items in cognition research [6,7,8]. However, the 2015–2020 Dietary Guidelines for Americans recommend a totality of foods for healthy living instead of focusing on individual food items [9]. As the nutrition interaction in the brain is complex, it is problematic to examine the independent effect of an individual food item on brain function when the individual food item is part of a broader diet [3]. In addition, the causes of cognitive decline are multifactorial and the impact of a single food item on cognitive decline might be too small to detect reliably [10]. As a result, researchers have shifted towards overall dietary approaches that may have interactive and cumulative effects on brain health improvement [5,11,12]. The 2015–2020 Dietary Guidelines for Americans (DGA) known as the Healthy Eating Index (HEI) [9] has been examined in relation to multiple health outcomes [13,14]. Still, critical gaps remain in studying the HEI and its role in CP [15,16].

In addition, we know very little about diet and cognition in older foreign-born adults, who comprise 13.9% or 7.3 million of the 52.5 million older adults in the US [17]. There is a dearth of literature about the intersection of diet, cognition, and nativity status as well as a lack of consensus, with some studies reporting higher CP in US-born individuals, in later years of life, when compared to foreign-born individuals [18,19,20,21]. At the population level, the higher CP in US-born older adults may be partly explained by better education; adequate nutrition in early years, adolescence and adult life; decreased exposure to air pollution; greater physical activity; and less psychosocial and acculturation stress [20,21,22,23]. By contrast, other studies report no difference in CP in early and late life between foreign-born individuals and their US-born counterparts [24]. These conflicting results may be due to methodological issues such as sample differences by gender, socioeconomic status and a lack of cultural measures in health behaviors such as diet [25]. Additionally, several studies have shown that disruption in dietary patterns results from immigration due to a lack of access to culturally specific food items and the consumption of calorie-dense foods widely available in the US context [26,27,28,29]. While most studies have adjusted for nativity status, we hypothesized that these substantive changes in diet across the course of life may result in differential effects on cognition later in life overall and by specific measures of cognitive performance.

This study examined the association between CP and adherence to the HEI by nativity status in a nationally representative sample of US- and foreign-born older adults of 60+ years of age. We hypothesize that meeting dietary adherence following the HEI-2015 will be associated with higher CP scores in older US-born adults than their foreign-born counterparts.

## 2. Materials and Methods

### 2.1. Data Source and Study Population

The National Health and Nutrition Examination Survey (NHANES) provides a nationally representative cross-sectional sample of the non-institutionalized US population. The details of the design of the NHANES are described elsewhere [30]. We combined the NHANES cycles from 2011 to 2014 and then merged them with the 2011–2014 Food Patterns Equivalents Database (FPED) to calculate the sample HEI-2015 dietary component and total diet scores [13].

Participants were excluded if they were <60 years of age (*n* = 13,852), had missing information on demographic characteristics (*n* = 187), dietary intake (*n* = 2), or digit symbol substitution test (DSST) (*n* = 288), Consortium to Establish a Registry for Alzheimer’s Disease (CERAD) battery immediate recall (*n* = 209) and delayed recall (*n* = 211), or animal fluency test (AFT) (*n* = 224). These exclusions yielded a complete data analytic sample of 3065 adults aged ≥60 years.

### 2.2. Cognitive Performance

CP tests in the NHANES were determined by experts on cognition, understanding that the selected measures would be succinct, suitable to diverse populations, and easy to administer. A mobile examination center (MEC) was the preferred setting for the administration of four in-person CP tests in older adults (aged ≥60 years). The NHANES includes select subdomains for working memory, language, processing speed, and executive functioning [31]. These domains were measured with the following: (i) the CERAD immediate learning test, (ii) the CERAD delayed recall, (iii) the AFT, and (iv) the DSST. The CERAD assessed the immediate learning and delayed recall of new verbal information [31]. The CERAD immediate recall consisted of three consecutive word-learning trials of 10-item word lists, and the 10-item word list was the same for each of the three consecutive trials. The scores from all three trial repetitions were summed, with higher scores indicating higher cognitive performance. The AFT measures verbal fluency, a component of executive function that was assessed by asking the participant to name as many animals as possible in one minute [31]. The DSST, a subtest of the Wechsler Adult Intelligence Scale, is a measure of processing speed, attention, and working memory that was assessed by asking the participants to match (pair) symbols to numbers in 2 min [31]. The CERAD delayed recall test provides a measure of delayed memory and was assessed by asking the participants to recall as many words as possible from the original ten words used in the CERAD test after the Animal Fluency and DSST tests were completed [31].

In this study, low CP was assessed as having scores in the lowest 25th percentile of each cognitive assessment. Participants with scores above the lowest 25th percentile were categorized as having adequate cognitive performance. This method has been applied previously in the analysis of national survey data [31,32,33]. We used this common epidemiologic technique of comparing the higher quartiles to lowest quartile because our study concentrated on respondents who performed adequately in comparison to other older adults who did not and, as a result, represented a wide range of community-dwelling individuals. However, this study did not include populations who may have a higher prevalence of cognitive impairment such as those living in nursing homes or other institutions. We did not perform sensitivity analysis for the CP tests, given that assessments using the same data and test-defined scoring in the lowest 25th percentile are the gold standard for low CP [31].

CP tests were administered in Spanish and English by trained bilingual interviewers and were available in a translated format for participants who spoke Korean, Vietnamese, or Chinese.

### 2.3. Diet Score (HEI-2015)

The Healthy Eating Index (HEI)-2015 was used to assess adherence to a healthy diet. To calculate the HEI-2015 score, dietary intake data were collected from the NHANES 24 h dietary recall data using the United States Department of Agriculture (USDA) automated multiple-pass method (AMPM) [34]. The HEI-2015 reflects basic food groups that can be applied to any culture or diverse group of population [35]. While mainly intended to reflect dietary guidance and adherence in the population of the United States, the HEI-2015 can be a useful tool for assessing diet among population subgroups. For example, it has been previously used for Mexican and Japanese populations [36,37]. Briefly, using the dietary recall data, detailed information about all foods and beverages a respondent consumed were captured. Individual food and nutrient data were measured using the USDA’s Food and Nutrient Database for Dietary Studies. Individual food data were then combined into 37 food groups (e.g., total intake of fruits, vegetables, or added sugars) known as Food Patterns Equivalents Database (FPED) to calculate the HEI-2015 components (food groups) [13,38]. Among the 13 components of the HEI-2015, 9 support diet adequacy (total fruit, whole fruit, total vegetables, greens and beans, whole grains, dairy, total protein foods, seafood and plant proteins, and fatty acids) and 4 should be consumed in moderation (refined grains, sodium, added sugar, and saturated fats) [14]. The sum of the components’ scores is the total HEI-2015 score ranging from 0 to 100, with a higher score indicating greater adherence to a healthy diet that aligns with the DGA [14]. For the purpose of our study, the total HEI-2015 scores were categorized into quintiles where the first quintile (Q1) represents the lowest diet quality and the last quintile (Q5) represents the highest diet quality. It was performed to best separate those with the highest dietary adherence (Q5) and lowest dietary adherence (Q1) [39]. This method has previously demonstrated evidence of construct validity, reliability, and criterion validity [40,41].

### 2.4. Demographic-, Socioeconomic-, and Health-Related Variables

We determined potential confounders a priori, which included self-reported age at time of interview, education, sex, race/ethnicity, and annual household income. We categorized age as 60 to 69, 70 to 79, and 80 years or more. Education was categorized as those with less than a high school education, high school, some college or associate degree (2 years of postsecondary education), and college degree or more; additionally, sex at birth (male and female); race/ethnicity (non-Hispanic White, non-Hispanic Black, Hispanic or Latino, non-Hispanic Asian); and annual household income level ($0 to $24,999, $25,000 to $74,999, and >$75,000) were categorized. We also included physical activity (the NHANES participants completed the internationally validated physical activity questionnaire or PAQ) categorized as ideal physical activity (PA) defined as ≥150 min of moderate-intensity activities per week or ≥75 min of vigorous-intensity activities per week, or an equivalent combination of both. Non-ideal PA was defined as 1–149 min of moderate-intensity activities per week and 1–74 min of vigorous activity or no PA [42]. Nativity status was categorized as participants born in the US and born outside the US or foreign-born participants.

### 2.5. Statistical Analysis

We generated weighted sample descriptive statistics and tested for differences by nativity status using χ^2^ tests (Table 1) and χ^2^ trend tests across the HEI-2015 diet quintiles by nativity status for each cognitive domain (Figure 1). The final analytical models examined the association between diet quintiles and CP stratified by nativity status (Table 2) and fit with separate multivariable logistic regression models for each CP domain. Model 1 evaluated crude or unadjusted associations between diet quintiles and each CP outcome; Model 2 adjusted for age, sex, race/ethnicity, income, education level; and Model-3 adjusted for Model 2 plus PA. All analyses incorporated stratum and sampling weights to account for the complex survey design. The statistical significance level was set at *p*-value < 0.05 (two-tailed). All analyses were performed using SAS 9.4 (SAS Institute Inc., Cary, NC, USA).

## 3. Results

Table 1 shows the demographic distribution of the sample by nativity status (*n* = 3065). The US-born participants had a significantly higher income (29.9% vs. 19.1%, *p*-value < 0.001) and relatively higher levels of education than foreign-born participants (some college: 32.8% vs. 17.0%, *p*-value < 0.001; college: 29.7% vs. 24.2%, *p*-value < 0.001). The US-born sample had higher percentages of non-Hispanic White (87.7% vs. 23.2%, *p*-value < 0.001) and non-Hispanic Black (9.0% vs. 7.0%, *p*-value < 0.001) older adults when compared to the foreign-born sample. The foreign-born sample had higher percentages of Hispanic or Latino (43.3% vs. 2.8%, *p*-value < 0.001) and non-Hispanic Asian (26.5% vs. 0.5%, *p*-value < 0.001) older adults than US-born older adults. The upper quintiles of the HEI-2015 had a slightly higher percentage of foreign-born participants compared to US-born participants (20.7% vs. 19.8%, 22.6% vs. 19.7%, 27.7% vs. 19.0%, *p*-value < 0.001).

The US-born participants had better performance scores in all four cognitive tests. Specifically, compared to foreign-born, US-born participants had adequate DSST (79.1% vs. 49.0%), AFT (80.8% vs. 54.7%), CERAD-immediate recall (78.8% vs. 65.6%), and CERAD-delayed recall (79.1% vs. 71.0%) cognitive performance.geriatrics-09-00025-t001_Table 1Table 1Sample characteristics of study participants by nativity status: NHANES 2011–2014 (*n* = 3065).CharacteristicsUS Born (*n* = 2322)Foreign Born (*n* = 743)*p*-Value
% (weighted frequency), *n*
Age group (years)

0.0960–6954.4 (1126)57.1 (477)70–7929.2 (714)31.0 (198)80 and over16.4 (482)11.9 (68)Sex

0.43Male46.2 (1131)44.3 (376)Female53.8 (1191)55.7 (367)Income

<0.001$0–$24,99923.3 (784)36.5 (277)$25,000–$74,99946.9 (971)44.3 (294)$75,000+29.9 (489)19.1 (124)Education level

<0.001<High school14.9 (530)41.1 (329)High school22.7 (577)17.7 (130)Some college 32.8 (706)17.0 (124)College degree+29.7 (508)24.2 (158)Race

<0.001Non-Hispanic White87.7 (1404)23.2 (59)Black9.0 (667)7.0 (73)Latino2.8 (181)43.3 (390)Asian0.5 (31)26.5 (211)LTPA

0.19Ideal29.6 (587)25.4 (184)Non-ideal70.4 (1735)74.6 (559)HEI quintile

<0.001Lowest quintile (Q1)20.5 (555)15.9 (115)2nd quintile (Q2)21.0 (482)13.1 (109)3rd quintile (Q3)19.8 (476)20.7 (158)4th quintile (Q4)19.7 (424)22.6 (161)Highest quintile (Q5)19.0 (384)27.7 (199)DSST

<0.001Low CP20.9 (709)51.0 (355)Adequate CP79.1 (1429)49.0 (285)AFT

<0.001Low CP19.2 (609)45.3 (278)Adequate CP80.8 (1580)54.7 (375)CERAD-IR

<0.001Low CP21.2 (611)34.4 (245)Adequate CP78.8 (1585)65.6 (416)CERAD-DR

<0.01Low CP20.9 (568)29 (188)Adequate CP79.1 (1629)71.0 (480)Note. College+, college degree or more education; LTPA, leisure-time physical activity; DSST, digit symbol substitution test; AFT, animal fluency test; CERAD-IR, CERAD immediate recall; CERAD-DR, CERAD delayed recall; CP, cognitive performance.

Figure 1A–D shows the prevalence of adequate CP (DSST, AFT, CERAD-immediate, and delayed recall, respectively) by HEI-2015 diet quintile and nativity status. Figure 1A,B shows significant increases in the prevalence of adequate DSST and adequate AFT with higher HEI-2015 quintiles in the US-born population compared to those in the lowest HEI-2015 quintile (*p* for trend = 0.001 for DSST and = 0.01 for AFT, respectively). In the US-born population, trends were not significant for CERAD-immediate (Figure 1C) and delayed recall (Figure 1D) (*p* for trend = 0.13 for immediate recall and = 0.15 for delayed recall, respectively). The prevalence of adequate DSST, AFT, and CERAD-immediate and -delayed recall also increased with higher scores on the HEI in foreign-born older adults. However, except for CERAD-immediate recall, none of the trend results reached statistical significance (*p* for trend = 0.21 for DSST, 0.50 for AFT, 0.003 for CERAD-immediate recall, and 0.83 for CERAD-delayed recall, respectively).Figure 1Prevalence of adequate (>25th percentile) DSST, AFT, CERAD-immediate recall, CERAD-delayed recall by HEI-2015 quintile and nativity status in individuals ≥60 years of age. (**A**) DSST; (**B**) AFT; (**C**) CERAD-immediate recall; (**D**) CERAD-delayed recall.
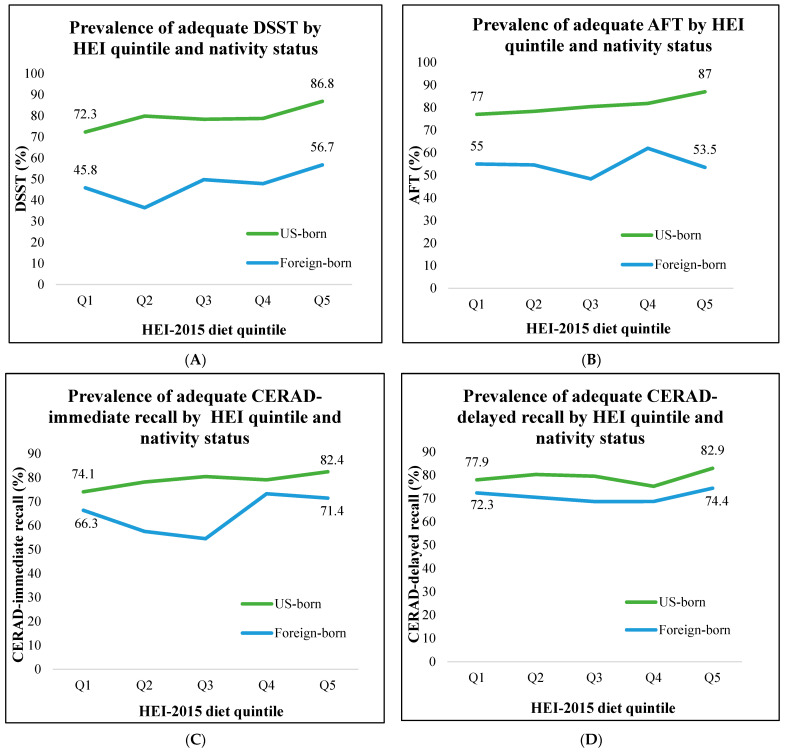


Table 2 presents stratified analyses between dietary adherence and each CP outcome by nativity status. Among participants who were US born, those in the highest diet quintile had significantly higher odds of adequate DSST compared to participants in the lowest diet quintile (Model 1: odds ratio (OR) = 2.51, 95% CI: 1.50–4.21, *p* = 0.001). This finding remained significant in the fully adjusted model (Model 3: OR = 1.95, 95% CI: 1.15–3.28, *p* = 0.01). Although not statistically significant in the full model, this pattern of association was also observed for AFT, CERAD immediate recall, and CERAD delayed recall. Among the foreign-born participants, unadjusted models showed a non-statistically significant trend for adequate DSST for the highest vs. lowest diet quintile (Model 1: OR = 1.55, 95% CI: 0.70–3.45, *p* = 0.27), as observed in the US-born group, but the direction of association was reversed in the fully adjusted model (Model 3: 0.61, 95% CI: 0.20–1.85, *p* = 0.37). The pattern, magnitude, and direction of associations in the foreign-born group were maintained for the three other cognitive performances (AFT, CERAD-immediate, and delayed recall). geriatrics-09-00025-t002_Table 2Table 2Odds ratios of adequate cognitive performance (DSST, AFT, CERAD-immediate recall, and CERAD-delayed recall) by diet quintile and nativity status, NHANES 2011–2014 (*n* = 3065).

US Born (*n* = 2322)Foreign Born (*n* = 743)Diet Q5 CP Tests*n*Model 1Model 2Model 3*n*Model 1Model 2Model 3


OR, (95% CI)
OR, (95% CI)HighestDSST3592.51 (1.50, 4.21) *2.14 (1.24, 3.68) *1.95 (1.15, 3.28) *1711.55 (0.70, 3.45)0.73 (0.23, 2.27)0.61 (0.20, 1.85)4th4011.42 (1.07, 1.88) *1.25 (0.83, 1.89)1.21 (0.81, 1.80)1421.08 (0.50, 2.36)0.65 (0.24, 1.71)0.57(0.21, 1.54)3rd4331.38 (1.01, 1.88) *1.32 (0.98, 1.79)1.29 (0.96, 1.74)1391.17 (0.58, 2.37)0.83 (0.31, 2.26)0.73 (0.28, 1.94)2nd4431.51 (1.02, 2.24) *1.51 (0.92, 2.48)1.48 (0.91, 2.42)940.68 (0.32, 1.44)0.70 (0.28, 1.73)0.64 (0.27, 1.53)Lowest501Ref.Ref.Ref.93Ref.Ref.Ref.Highest AFT3612.01 (1.25, 3.21) *1.56 (0.95, 2.55)1.42 (0.88, 2.30)1700.94 (0.49, 1.81)0.97 (0.43, 2.15)0.91 (0.41, 2.01)4th4041.34 (1.01, 1.79) *1.17 (0.83, 1.66)1.13 (0.80, 1.60)1431.34 (0.63, 2.81)1.75 (0.68, 4.53)1.68 (0.66, 4.31)3rd4451.23 (0.83, 1.80)1.15 (0.73, 1.82)1.13 (0.72, 1.77)1440.77 (0.38, 1.56)0.97 (0.37, 2.51)0.92 (0.36, 2.39)2nd4581.08 (0.72, 1.62)0.89 (0.57, 1.41)0.88 (0.55, 1.39)940.99 (0.43, 2.25)1.14 (0.45, 2.90)1.09 (0.44, 2.75)Lowest520Ref.Ref.Ref.101Ref.Ref.Ref.Highest CERAD-IR3621.65 (1.06, 2.55) *1.43 (0.92, 2.21)1.42 (0.90, 2.24)1731.27 (0.70, 2.29)0.89 (0.54, 1.49)0.89 (0.50, 1.58)4th4031.33 (0.96, 1.83)1.25 (0.86, 1.84)1.25 (0.85, 1.84)1431.39 (0.81, 2.40)1.15 (0.65, 2.04)1.15 (0.64, 2.08)3rd4451.44 (1.02, 2.03) *1.37 (0.95, 1.98)1.37 (0.94, 2.00)1480.61 (0.35, 1.06)0.57 (0.27, 1.22)0.57 (0.26, 1.28)2nd4591.25 (0.89, 1.76)1.16 (0.77, 1.76)1.16 (0.77, 1.75)970.69 (0.40, 1.17)0.88 (0.44, 1.74)0.88 (0.41, 1.85)Lowest526Ref.Ref.Ref.99Ref.Ref.Ref.HighestCERAD-DR3631.37 (0.89, 2.12)1.32 (0.80, 2.17)1.26 (0.75, 2.11)1721.10 (0.60, 2.06)0.78 (0.43, 1.39)0.77 (0.42, 1.38)4th4030.86 (0.69, 1.06)0.79 (0.57, 1.10)0.78 (0.56, 1.08)1430.84 (0.40, 1.75)0.64 (0.30, 1.38)0.63 (0.28, 1.41)3rd4451.10 (0.75, 1.60)1.03 (0.68, 1.55)1.01 (0.67, 1.53)1470.84 (0.39, 1.78)0.71 (0.37, 1.34)0.70(0.36, 1.37)2nd4591.15 (0.79, 1.68)1.11 (0.74, 1.65)1.09 (0.73, 1.64)960.91 (0.53, 1.57)0.68 (0.32, 1.41)0.67 (0.32, 1.41)Lowest526Ref.Ref.Ref.99Ref.Ref.Ref.Note. * *p*-value < 0.05 refers to comparison between upper four quintiles with the lowest quintile when stratifying the association between diet quintiles and CP by nativity status; CP tests, cognitive performance tests (outcome); CERAD-IR, CERAD immediate recall; CERAD-DR, CERAD delayed recall; DSST, digit symbol substitution test: outcome (score > 25th percentile); AFT, animal fluency test: outcome (score > 25th percentile); CERAD-immediate recall: outcome (score > 25th percentile); CERAD-delayed recall: outcome (score > 25th percentile); diet Q5, diet quintile = main predictor; nativity status (US born vs. foreign born) = stratifier; OR = odds ratio, CI = confidence interval; Model 1—crude or unadjusted model with diet quintiles as the main predictor; Model 2—adjusted for age, sex, educational level, income; Model 3—adjusted for Model 2 plus leisure time PA (LTPA).

## 4. Discussion

Older adults represent a large segment of the US population and preventive strategies are needed to reduce the decline in cognitive function in this population. Our study found that US-born participants in the highest HEA-2015 quintile had greater odds of adequate DSST scores compared to the lowest diet quintile. Even though it was non-statistically significant, this pattern of association was also observed for AFT, CERAD-immediate recall, and CERAD-delayed recall. In contrast, for foreign-born individuals, participants in the highest quintile had a lower odds of adequate cognitive test scores, but the results were statistically non-significant.

Unlike most of the previous studies which have concentrated on specific dietary patterns (i.e., the Mediterranean diet pattern (MeDi) and the Dietary Approaches to Stop Hypertension (DASH) diet pattern), our study adds to the field of diet and cognition among racially/ethnically diverse populations by looking at multiple dietary components. Prior work using the Healthy Eating Index (e.g., HEI-2005, HEI-2010) has shown associations with CP in African Americans and Puerto Ricans [43], yet no study has examined these differences by nativity status. One investigation applying the HEI-2015 showed higher adherence to dietary guidelines was associated with better CP, but the study did not examine differences by nativity [44]. Our findings showing a positive association for US-born adults point to potentially important differences by nativity status that need further exploration and add more nuance to the mixed evidence on the positive association between adherence to a healthy diet and adequate CP [45,46].

Differences in the association between dietary adherence and CP by nativity status showed a positive gradient in dietary quintiles in US-born older adults and an inverse association (non-significant) for foreign-born adults. Previous studies have shown that US-born people are more likely to have higher CP than people who are foreign born [47]. This higher CP in US-born people can be partially explained by several factors such as quality of education; wealth; adequate nutrition in early and adult life; less exposure to disease pollutants and smoking; more physical mobility; and fewer physical-, social-, and migration-related stressors [19,20,21]. This explanation aligns with our study results characterized by significantly higher incomes and relatively higher levels of education in the US-born participants than in the foreign-born participants. Another potential explanation suggests that foods rich in micronutrients, dietary fiber, omega-3 fatty acids, vitamins C and E, folates, and other carotenoids may alter inflammatory pathways and exert anti-inflammatory and antioxidant effects via reductions in oxidative stress and lipid peroxidation [48]. Furthermore, a high-quality diet may modify the risk of other vascular diseases (e.g., stroke and cardiovascular events) and delay cognitive decline by reducing microbleeds and subsequent ischemia in the brain [49]. Taken together, greater adherence to a high-quality diet across the lifespan could be associated with a lower risk of cognitive decline in older adults [50]. Future longitudinal studies that include more racially/ethnically diverse older adults as well as other axes of social stratification are needed. Finally, our study findings show significant positive gradients in dietary quintiles for US-born adults in the case of the DSST, suggesting that adherence to the HEI-2015 may have better benefits for processing speed, attention, and working memory. Previous studies have similarly shown varying results when testing multiple dietary indices and domains of CP [51,52]. Given the cross-sectional nature of our study, interdisciplinary, longitudinal research is needed to determine the causal link between healthy diet indices and their differential effect on specific cognitive measures.

The dietary habits of the foreign-born population in the US also depend on food availability, cultural values, meal frequency, and snacking. Foreign-born individuals may have work responsibilities, work hours, and higher work stress levels which leave them with less time to prepare nutrient-rich meals and, as a result, they may eat lower-quality, more affordable foods [29]. However, some studies have also found no significant difference in CP by nativity status, or have shown that CP is lower among those who are US-born as compared to the foreign born [24]. Our results were clearly less consistent for foreign-born participants and showed healthy diet consumption to be associated with poorer CP. Future studies with larger samples of foreign-born participants are needed to better assess the complex relationship between migration factors (e.g., age at migration, place of residence, food insecurity in childhood vs. adult life), diet, and cognitive performance in older adults.

The limitations of our study include the cross-sectional approach, which limits the causal inference of our associations. Prospective study is needed to further validate this study. Secondly, the CP tests evaluated in our study do not cover all domains of cognition. Thirdly, our finding of non-significant associations between diet and cognition for the DSST, AFT and CERAD-immediate and delayed recall may be related to the lack of culturally specific cut points for foreign-born older adults. Despite following the guidelines from prior studies to classify ‘adequate’ CP using a cut off score (>25th percentile), we still found mixed findings [32,33,53]. The non-significant findings may also be due to our smaller sample size in the foreign-born group. Fourthly, we did not use time in the US as a covariate and did not evaluate if longer stays in the US increase or decrease cognitive performance in the foreign-born group. However, previous studies report the effect of migration on CP as mixed and somewhat unclear [18]. Finally, the study population included a racially/ethnically diverse group and although interpreters were used, there may have been an effect from the language used or cultural factors that influenced measured score [31].

Despite these potential limitations, our study has several strengths. This study explored the association between the HEI-2015 and CP in older adults by nativity status. Moreover, we used a large sample of older adults representing community-dwelling, diverse racial and ethnic groups in the United States. We also applied cognitive assessments and diet quality measures using detailed and widely accepted validated tools in the field.

## 5. Conclusions

Our study adds compelling evidence on the role of diet and cognition in a population-based sample of US-born and foreign-born older adults. Specifically, we found that DSST, a marker of attention, processing speed, and executive function is higher among US-born older adults consuming a higher quality diet. Future research should consider the role of the timing of the healthy diet in delaying or preventing cognitive decline and factors that contribute to worse outcomes for the foreign-born population.

## Data Availability

Publicly available datasets were analyzed in this study. All data and materials are publicly available at the National Center for Health Statistics (NCHS) website: https://www.cdc.gov/nchs/nhanes/index.htm (accessed on 18 January 2021).

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
