# Peer review of "Dietary Adherence and Cognitive Performance in Older Adults by Nativity Status: Results from the National Health and Nutrition Examination Survey (NHANES), 2011–2014"

_geriatrics, 2024, doi:10.3390/geriatrics9020025_

Round 1

Reviewer 1 Report

Comments and Suggestions for Authors

Thank you so much for the opportunity for me to review this article investigating the association between dietary quality, cognitive function, and nativity among U.S. older adults using a national representative sample. The authors found that higher dietary quality was associated with higher odds of having adequate cognitive function. The association was not statistically significant for the non-U.S. born individuals.

I have some questions and comments regarding this manuscript. Please find them listed below.

Introduction

1. Is there any evidence on verifying the diet quality scale among diverse population? Does the scale consider to be culturally competent? I brought this issue since that dietary adherence advice sometimes did not consider cross-cultural differences. An individual with different dietary culture may eat healthily but is not captured in a scale.

2. What are the hypotheses of the study?

3. I would recommend the authors gave rationale on why they incorporated nativity into the analyses. It is not very clear from reading the introduction to see what the authors thought about the role of nativity on the association between dietary adherence and cognitive function.

Methods

1. It is not very clear to the reader about the timeframe of when exposure, outcome, and covariates were measured. It seems that this is a cross-sectional study and I would expect the authors to be more clearer about this.

2. How were the cognitive function scores measured in this study? Were they measured by trained interviewers?

3. Why did the authors use 25 percentile of cognitive performance as the outcome? Did they do any sensitivity analyses on using different outcomes of cognitive function score?

4. The authors used quantiles for the healthy eating index, I would recommend them provide a rationale on choosing quantiles as the measure.

5. Besides age, were there any inclusion/exclusion criteria for the study participant recruitment?

6. For the modeling, the authors should mention what statistical models they used for analyses.

Conclusion

1. It appears that U.S. born and foreign-born participants differed across sociodemographic characteristics, such as educational attainment and socioeconomic status, those could be important factors of cognitive performance. I would recommend the authors comment on this sociodemographic differences between the two samples.

2. For the limitation, the authors should also address the limitation of using cross-sectional data and therefore limited causal inference could be derived from the analysis.

3. The authors could mention some implications and lesson learned from this analysis.

Reviewer 2 Report

Comments and Suggestions for Authors

I believe this paper is quite interesting. However, the lack of clarity in articulating the research question and detailing the research methods makes it difficult to assess the results, discussions, and conclusions.

  1. The last paragraph of the introduction creates confusion about the research question. In particular, it's unclear how the comparison between the US-born group and the non-US-born group is intended. In addition, the research hypothesis is ambiguous, so I ask that additional information be added to the research question, taking into account other feedback.
  2. Please include a description of the model used for the analysis in Table 2.
  3. In fig1, the horizontal axis is the value of HEI-2015 and in the analysis of Table 2, the explanatory variable reads as the value of CP, which is confusing; please correct fig1 so that the horizontal axis is CP in fig1.
  4. Please explain the dependent variable in Table 2.
  5. What is the reason for stratifying US and non-US birth groups in Table 2? If the hypothesis is that 'US birth groups with high HEI have the highest CP and non-US birth groups with low HEI have the lowest CP', could it be possible to consider analysing the data in one model without stratification?
  6. In the final model, a significant association was observed only in the US-born group between HEI-2015 and DSST in the highest quartile. Why is that the case? Please add discussions on the reasons for the lack of significant differences in other cognitive performances (CP) and in the non-US-born group

Additional details:

  1. Line 30: It mentions "cognitive funciton(CP)." Would it be acceptable to use "Cognitive performance" consistently?
  2. Line 70: Why was it explicitly mentioned to exclude pregnant women? (Pregnant women over the age of 60 are not usually found.)
  3. Line 119: What does "some college" specifically refer to? Please explain to readers abroad who don't understand the American school system.
  4. Reference 25: Could you please add the DOI?
Comments on the Quality of English Language

The last paragraph of the introduction might pose challenges for non-native speakers, so consider shaping it in a style akin to an example from a writing textbook.

Round 2

Reviewer 2 Report

Comments and Suggestions for Authors

Thank you for your response to my comments.

I think they are generally good. There seem to be no problems with the research methodology, but please discuss some of the sentences with your academic editor.

Reply to reply to comment 1

Thank you for your reply and for adding the sentence. However, the hypothesis sentence is difficult to understand for someone like me who is not a native English speaker.

Is it correct that the hypotheses you added are 

1) CP will be lower if the diet conforms to HEI-2015

2) if you compare people from non-US and US backgrounds, CP will be lower overall for people from non-US backgrounds if they follow a diet that conforms to HEI-2015 compared to people from the US?

For reference, I have also translated it at DeepL, and it translates similarly to my translation. Please discuss with your academic editor how to write it.

I also think it would be better not to emphasise too much that it is associated with 'low CP' because I felt that emphasising 'low CP' would confuse the reader, as the intended outcome of the analysis that follows is that 'CP is not in the bottom 25%' (i.e. no decline).

Response to reply to comment 2

Thank you for the additional information.

I understand.

Response to reply to comment 3

Thank you for the additional explanation.

I understand.

Response to reply to comment 4

Thank you for the additional information.

I understand.

Response to reply to comment 5

Thank you for the additional explanation and the addition to the text.

I understand.

Response to reply to comment 6

Thank you for the additional explanation.

I understand.

In reply to your response to my comments on additional details

Reply to 1. and 2.

Thanks for the correction. I understand.

Reply to 3.

Thank you for the addition. I understand.

Response to 4. 

I understand that there is no DOI. Please discuss with the academic editor how to describe the details.

Comments on the Quality of English Language

Thanks for the correction. There are no issues other than what I described in my response to your reply to comment 1.
